# An integrated multimodal model of alcohol use disorder generated by data-driven causal discovery analysis

Eric Rawls [1✉], Erich Kummerfeld[2] & Anna Zilverstand[1]

Alcohol use disorder (AUD) has high prevalence and adverse societal impacts, but our understanding of the factors driving AUD is hampered by a lack of studies that describe the complex neurobehavioral mechanisms driving AUD. We analyzed causal pathways to AUD severity using Causal Discovery Analysis (CDA) with data from the Human Connectome Project (HCP; n = 926 [54% female], 22% AUD [37% female]). We applied exploratory factor analysis to parse the wide HCP phenotypic space (100 measures) into 18 underlying domains, and we assessed functional connectivity within 12 resting-state brain networks. We then employed data-driven CDA to generate a causal model relating phenotypic factors, fMRI network connectivity, and AUD symptom severity, which highlighted a limited set of causes of AUD. The model proposed a hierarchy with causal influence propagating from brain connectivity to cognition (fluid/crystalized cognition, language/math ability, & working memory) to social (agreeableness/social support) to affective/psychiatric function (negative affect, low conscientiousness/attention, externalizing symptoms) and ultimately AUD severity. Our data-driven model confirmed hypothesized influences of cognitive and affective factors on AUD, while underscoring that addiction models need to be expanded to highlight the importance of social factors, amongst others.

[1] Department of Psychiatry and Behavioral Sciences, University of Minnesota, Minneapolis, MN, USA. [2] Institute for Health Informatics, University of Minnesota, Minneapolis, MN, USA. ✉email: erawls89@gmail.com

Lifetime incidence of alcohol use disorder (AUD) is as high as 29–30%[1,2], with alcohol use being a leading cause of death [3 million worldwide in 2016 or 5.3% of all deaths worldwide[3]]. Success rates for quitting drinking in AUD are low [30–40%[4,5]], which has been attributed to the multi-causality of the mechanisms underlying AUD and the need for more targeted treatments[6]. However, the development of targeted interventions for AUD is hampered by a lack of studies investigating multifactorial mechanisms driving AUD.

Early theories of addiction maintenance proposed single key mechanisms, such as allostasis[7], or hedonic signaling[8]. These early theories have given way to multifactorial models of addiction, such as the "three-stage cycle" model[9], which proposes that negative affect, incentive salience, and executive function are functional domains involved in addiction. There is a great deal of empirical support for the involvement of these three domains. The three domains have been mapped onto corresponding personality profiles that confer addiction risk[10] and have been used to develop a set of proposed neuroclinical assessment tools[11,12] that were successfully applied to AUD[13]. However, a three-domain model is far from encompassing the entire phenotypic space that contributes to AUD. The NIMH RDoC[14,15] proposed 23 functional domains underlying psychopathology, recognizing a need for multivariate models that incorporate increasing knowledge of the many functional domains contributing to psychiatric dysfunction. A recent consensus paper on a multivariate assessment approach for addiction identified another seven "addiction-specific" domains in addition to the RDoC domains[16]. Critically, in all of these approaches, the proposed functional domains were identified by expert consensus and therefore might not exactly match the true underlying domains that exist in the data. For example, in an exploratory analysis of a large public dataset, Van Dam et al.[17] derived seven phenotypic factors that only partially mapped onto RDoC domains, but predicted psychiatric distress. A more recent addiction theory[18] identified ten domains contributing to maladaptive decision-making in addiction. A systematic review of neuroimaging studies in addicted populations implicated the involvement of at least six different neurobiological mechanisms in AUD[19]. These recent developments underscore the need for data-driven, multivariate analysis methods capable of fully examining and describing the large phenotypic space underlying addiction, if we are to understand the central question of how multi-causal factors underlie the maintenance and escalation of alcohol use.

In the current study, we leveraged the deep behavioral and psychiatric phenotyping[20] and high-resolution neuroimaging data[21] from the Human Connectome Project (HCP)[22]. Using data from nearly 1000 participants, we first derived a set of data-driven domains underlying the full range of phenotypic functioning measured in the HCP dataset. We extracted whole-brain connectivity metrics from 12 data-derived resting-state functional magnetic resonance imaging (fMRI) networks[23] to measure individual neurobiological differences. To examine the relationships between fMRI network connectivity, phenotypic domains, and AUD symptom severity, we applied Causal Discovery Analysis (CDA), a class of machine learning techniques that learns causal models from input data. These methods search the enormous set of possible structural models and return a graph representing estimated causal relationships in the data. The particular method we applied, Greedy Fast Causal Inference (GFCI)[24], uses conditional dependence relations to discover when unmeasured variables confound the relationships between measured variables, making this method particularly powerful for real-world data sets that cannot possibly capture every variable of interest. By (1) deriving data-driven domains encompassing the whole phenotypic space measured in HCP, (2) extracting whole-

brain network connectivity profiles, and (3) applying CDA to the resulting phenotypic and neurobiological domains, we generated an integrated, multimodal causal model of neurobehavioral factors contributing to AUD symptom severity.

## Results

**Exploratory factor analysis: decomposing the phenotypic space measured in the HCP.** To reduce the phenotypic space measured in the HCP to a set of underlying domains, we conducted an exploratory factor analysis (EFA) in the entire HCP sample that had complete phenotypic data ($n = 933$, 53.5% females). Based on the results of Monte Carlo simulation we extracted 18 factors ($p < 0.05$) from the 100-variable phenotypic space measured in the HCP dataset, which collectively accounted for 47% of common variance. Results of the Monte Carlo permutation test for eigenvalue significance are presented in Supplementary Fig. S1 (eigenvalue significance), Supplementary Data 1 (observed, random, and resampled eigenvalues), and Supplementary Table S1 (percent variance explained per factor, eigenvalues, and cumulative variance). EFA model fit indices indicated good factor separation (RMSEA = 0.03, Tucker–Lewis Index = 0.86).

Factors, in order of common variance accounted for, were associated with: (1) Somaticism (high DSM/ASR somaticism, high DSM depression, low PSQI sleep quality), (2) Fluid Cognition (high Raven's progressive matrices performance), (3) Internalizing (high DSM/ASR anxiety, high DSM depression, high NEO-FFI neuroticism), (4) Gambling Task Reaction Time (slow gambling task reaction time), (5) Conscientiousness/Attention (low DSM attention deficit hyperactivity disorder, low ASR attention problems, and high NEO-FFI conscientiousness), (6) Visuospatial Processing (high Penn short line orientation task performance), (7) Social Support (high NIH toolbox friendship, low loneliness, low perceived rejection and perceived hostility, high emotional and instrumental support), (8) Processing Speed (high NIH Toolbox Flanker Total Score, fast fMRI emotion task RT), (9) Externalizing (high ASR aggression and rule-breaking, high DSM antisocial, high NIH toolbox aggression), (10) Social Withdrawal (high ASR withdrawal, high DSM avoidance, low NEO-FFI extraversion), (11) Language Task Performance (high fMRI language task story average difficulty, and high math problem accuracy), (12) Relational Task Reaction Time (slow fMRI relational task reaction time [RT]), (13) Delay Discounting (high delay discounting AUC for $200 and $40k), (14) Working memory (fMRI N-Back task fast reaction time [RT] and high accuracy), (15) Negative Affect (high NIH toolbox anger, fear, sadness and stress), (16) Crystalized IQ (high NIH toolbox English reading and picture vocabulary, high education, and high NEO-FFI openness), (17) Positive Affect (high NIH toolbox life satisfaction, positive affect, and meaning and purpose, and NEO-FFI extraversion), and (18) Agreeableness (low aggression and high NEO-FFI agreeableness). Factor loadings for each item are available in Supplementary Data 2 (raw variables) and Supplementary Data 3 (standardized variables), and the factor correlation structure is available in Supplementary Data 4.

## Causal discovery of the neurobehavioral underpinnings of AUD

*Model fit and quality metrics.* To build a multidomain model of the causal neurobehavioral underpinnings of AUD, we used a recently developed causal discovery machine learning algorithm called GFCI[24], which is particularly powerful for use with observational data because it has the capacity to determine when causal relationships are impacted by unobserved confounding variables. We submitted the 18 factors extracted by EFA,

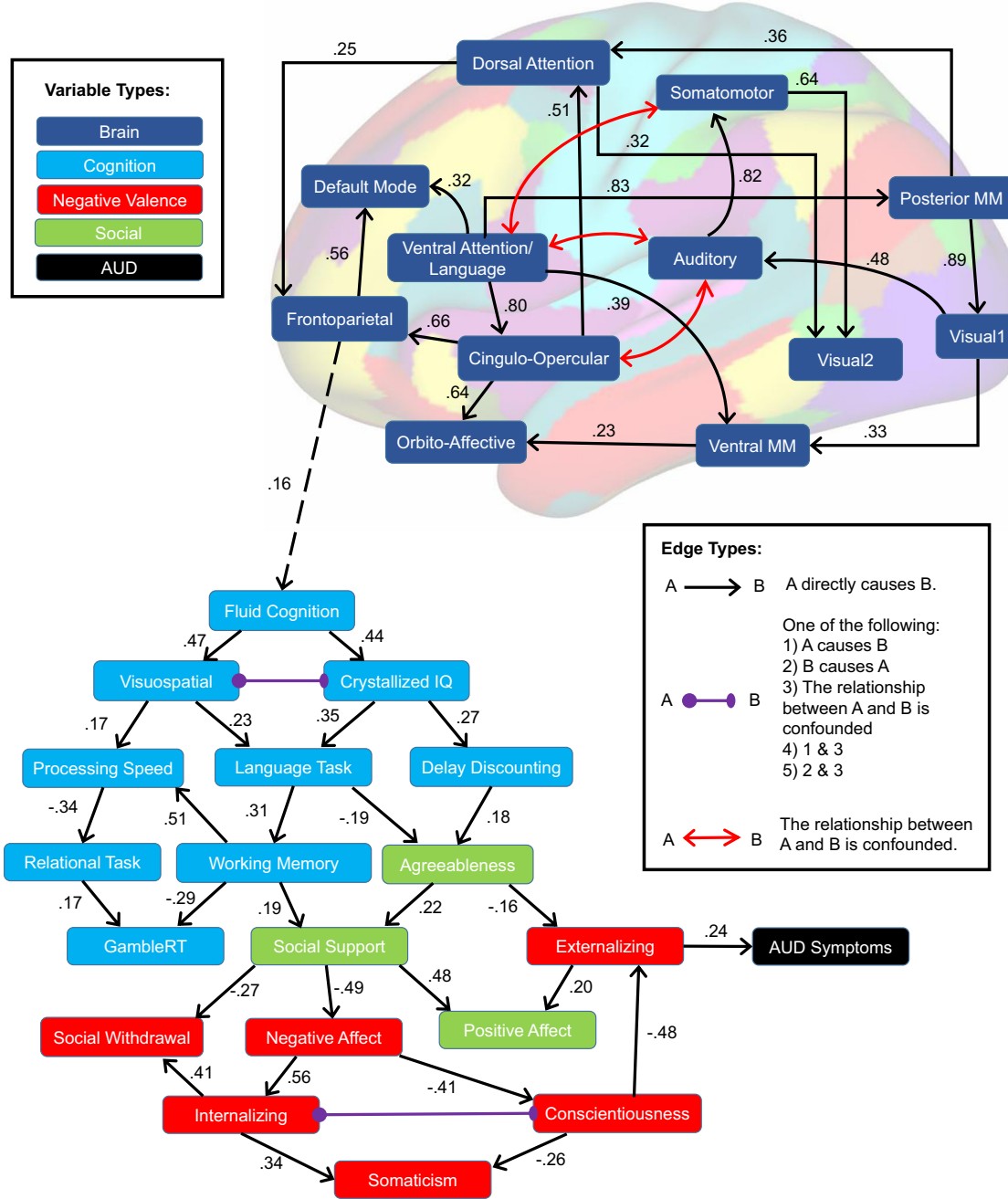

**Fig. 1 Causal discovery of the neurobehavioral underpinnings of alcohol use disorder.** Causal discovery analysis of the neurobehavioral determinants of AUD symptom severity in the HCP dataset was done using Greedy Fast Causal Inference (GFCI). GFCI returns a partial ancestral graph (PAG) depicting causal relationships between a set of variables, while assessing for unmeasured third variables in relationships (confounders). Standardized edge weights recovered via structural equation modeling (SEM) are displayed in text next to each edge in the graph. The overall SEM fit was good, RMSEA = 0.06, Tucker-Lewis Index = 0.91.

within-network brain resting-state fMRI connectivity from 12 previously defined resting-state networks[23], and a measure of AUD severity (AUD symptom count) to a unified causal modeling framework using GFCI. The output of GFCI is presented in Fig. 1.

We extracted measures of model fit and effect sizes for causal relationships using structural equation modeling (SEM). The SEM fit to this model indicated a good fit, RMSEA = 0.06, Tucker-Lewis Index = 0.91, and every edge in the causal model was significant in the corresponding SEM ($p < 0.001$). Recovered edge weights from SEM were presented overlaid on the GFCI graph. Stability testing (jackknife analysis) demonstrated the high

stability of the model (Supplementary Fig. S2 and Supplementary Data 5). Furthermore, the model was highly replicable when using a separately defined brain network parcellation (RMSEA = 0.06, Tucker–Lewis Index = 0.87, every edge $p < 0.001$) (Supplementary Fig. S3). As such, the results described here do not appear to depend on the specific parcellation used to measure fMRI resting-state network connectivity.

*Causal model results interpretation.* First, we found that brain network connectivity measures and phenotypic factors largely separated into two interconnected separate clusters. The brain

network subgraph indicated several salient points. We found that high connectivity within the ventral attention/language network during rest caused high connectivity within default mode (DMN), cingulo-opercular, and multimodal sensory association networks —networks that play a central role in self-reflective brain processes. We also found effects indicating causal influences of cingulo-opercular connectivity on attentional processing (cingulo-opercular → dorsal attention). Finally, we found converging causal influences onto the frontoparietal network (cingulo-opercular/posterior multimodal → frontoparietal, dorsal attention → frontoparietal).

Brain connectivity intersected with behavioral phenotypic variables in a link between high frontoparietal connectivity and high Fluid Cognition. From this point, causal influences propagated from Fluid Cognition to Visuospatial Processing and Crystalized IQ, replicating a well-studied effect that individuals high in fluid cognitive ability will also be high in crystalized intelligence[25,26]. From there causal influences proceeded to more specific cognitive measures, including Working Memory, Language Task Performance (verbal and math ability), and Delay Discounting. We found a direct link between Crystalized IQ and Delay Discounting, such that individuals higher in Crystalized IQ also exhibited lower (less impulsive) discounting rates. These cognitive measures were then in turn causally linked to affective, social, and psychiatric factors. High Language Task (verbal and math) performance and less impulsive Delay Discounting caused Agreeableness. Low Working Memory performance, and low Agreeableness caused lowered Social Support, and decreased Social Support contributed to increased Negative Affect, increased Social Withdrawal, and decreased Positive Affect. High Negative Affect in turn contributed to higher Internalizing symptoms, and lower Conscientiousness/Attention. Low Conscientiousness/Attention, and low Agreeableness caused high Externalizing psychopathology, while high Externalizing psychopathology directly caused increased AUD symptom severity.

Previous hypotheses have particularly focused on the influences of negative affect, incentive salience, and executive function in AUD. Our results support a causal role for cognitive and affective influences on AUD, while expanding our understanding of the complex multifactorial space contributing to AUD.

## Discussion

Early addiction models posited that addiction was due to single key mechanisms[7,8], but modern addiction models have begun to emphasize the multifactorial mechanisms underlying addiction[9,11,16,18,19]. In this study, we used a data-driven approach to characterize phenotypic domains in a large community sample, and examined whole-brain network connectivity at a large scale using a data-driven network analysis and parcellation[23]. We then modeled large-scale brain and behavioral influences on AUD symptom severity using CDA. Our results shed light on the relationship between brain network connectivity and phenotypic domains in general, as well as providing specific information on how brain and behavioral factors contribute to the severity of AUD, and which could be targeted in treatment.

Our analysis significantly expands the multifactorial space of current addiction models. Our factor analysis uncovered a variety of factors that map relatively well onto domains elaborated in RDoC (Table 1). For example, we found factors that mapped well onto aspects of the RDoC Cognitive Systems domain, the RDoC Negative Valence Systems domain, and the RDoC Social domain. To assist in interpreting the large-scale domains that our data-driven factors mapped onto, we grouped factors based on their correlations. Interestingly, we found that the Conscientiousness/ Attention and the Social Withdrawal factors correlated with other factors in the Negative Valence Systems domain, rather than the RDoC-assigned grouping of these factors (Cognitive Systems: Attention and Social Systems: Affiliation & Attachment, respectively). A previous review found that inattention and anxiety are tightly linked[27], but our results provide evidence of the direct link between inattention and negative affect. We also found that the Delay Discounting factor, while considered part of the RDoC Positive Valence System: Reward Valuation subconstruct[28], correlated instead with Cognitive Systems factors, suggesting delay discounting is more related to Cognitive Control/Impulsivity domains than to Reward Valuation[29–31].

Here we summarize several key points from our mapping of causal influences on AUD symptom severity onto the RDoC framework. First, our analysis uncovered strong evidence for the direct causal effect of the Negative Valence domain in AUD. This specifically included a causal influence of the Negative Affect factor on AUD, mediated through Conscientiousness/Attention (which correlated with other Negative Valence Systems factors). While many neurobiological models of addiction agree on the importance of negative affect in AUD[9,19], this is not unanimously agreed upon by experts in the addiction sciences[16]. The presented empirical data hence provides important empirical evidence implicating the broader Negative Affect Domain (as defined in RDoC) as an important treatment target in AUD.

Second, our analysis uncovered strong evidence for a mediating/buffering role of the Social Systems domain in AUD. Low Social Support and low Agreeableness were indirect causes of AUD severity and fully mediated the effect of cognition on the negative valence domain, providing strong empirical evidence that addiction models should incorporate measures of social function[32,33]. Epidemiological research has established a solid link between social affiliation and drug addiction[34], and increased social affiliation is associated with decreased risk of relapse in drug users who are seeking treatment[35]. Despite the considerable evidence research has uncovered for the importance of social affiliation as a protective factor in addiction[36], current neurobiological models of addiction generally fail to consider social factors[32] and their close relationship to cognitive/affective factors.

We generally found weak evidence for the involvement of the Positive Valence Systems domain in this analysis, although this is likely due to a limitation of the dataset employed. Positive Valence Systems subdomains, including reward-based domains that are particularly important in addiction[16] are relatively neglected in the HCP dataset[37]. We did find a causal influence of Delay Discounting on AUD severity, but as noted previously the Delay Discounting factor correlated with other Cognitive factors, suggesting that Delay Discounting (as measured in the HCP study) might be related to Cognitive Control/Impulsivity more so than Reward Valuation[29–31].

Finally, our analysis provides strong evidence that prefrontal cortex (PFC) brain networks, and associated cognitive factors, are situated at the top of the causal hierarchy of influences on AUD severity. The role of PFC and associated high-level cognitive factors in addiction is often referenced and is a part of major current theories of addiction[19,38], but our results are among the first to empirically demonstrate this hierarchical influence on AUD. Importantly, our causal model indicates that cognitive influences on AUD severity may extend far beyond the traditional consensus that inhibitory control is the most important cognitive influence on addiction[16,38,39], as we also highlighted influences of fluid and Crystallized IQ, working memory, and language/math ability on AUD.

Our results also shed light on theorized externalizing and internalizing pathways to AUD. Previous research has shown that externalizing symptomatology predicts AUD in young adult

**Table 1 Discovered factors (EFA) using 100 phenotypic measures.**

| Domains | Factors grouped according to factor correlations[a] | RDoC subdomains[b] |
|---|---|---|
| Negative Valence | Externalizing (high ASR aggression and rule-breaking, high DSM antisocial, high NIH aggression) | Frustrative Non-reward |
| | Conscientiousness/Attention (low DSM ADHD, low ASR attention problems, and high NEO-FFI conscientiousness) | *Attention*[c] |
| | Somaticism (high DSM/ASR somaticism, high DSM depression, low PSQI sleep quality) | Sustained threat |
| | Internalizing (high DSM/ASR anxiety, high DSM depression, high NEO-FFI neuroticism) | Potential threat, Sustained threat |
| | Negative Affect (high NIH anger, fear, sadness and stress) | Acute threat, loss, sustained threat |
| | Social Withdrawal (high ASR withdrawal, high DSM avoidance, low NEO-FFI extraversion) | *Affiliation & Attachment*[c] |
| Cognition | Visuospatial Processing (high Penn short line orientation task performance) | Visual |
| | Delay Discounting (high delay discounting AUC for $200 and $40k) | *Reward valuation*[c] |
| | Language Task Performance (high fMRI language task story average difficulty, and high math problem accuracy) | Language behavior |
| | Crystalized IQ (high NIH English reading and picture vocabulary, high education, and high NEO-FFI openness) | Declarative memory |
| | Fluid Cognition (high Raven's progressive matrices performance) | Working memory |
| | Gambling Task Reaction Time (slow gambling task reaction time) | |
| | Working memory (fMRI N-Back task fast reaction time (RT) and high accuracy, fast Penn word memory RT) | Declarative/working memory |
| | Processing speed (high NIH flanker total score, fast fMRI emotion task RT) | |
| | Relational Task Reaction Time (slow fMRI relational task RT) | |
| Social | Social Support (high NIH friendship, low loneliness, low perceived rejection and perceived hostility, high emotional and instrumental support) | Affiliation & Attachment |
| | Positive Affect (high NIH life satisfaction, positive affect, and meaning and purpose, and NEO-FFI extraversion) | Perception and Self |
| | Agreeableness (low aggression and high NEO-FFI agreeableness) | Affiliation & Attachment |

[a]Factors are grouped according to correlations between the factors.
[b]The right column indicates the RDoC domain each factor most closely approximated.
[c]Factors whose correlation structure did not match the RDoC domain assignment for that factor ($n = 3$) are displayed in italics.

samples[40]. Our data-driven causal model revealed that externalizing fully mediated the impact of all other (measured) causes of AUD; that is, AUD is unrelated to other phenotypic or brain network factors when externalizing is controlled for. Note that our externalizing factor consisted of ASR rule-breaking, aggression, and antisocial scales, and NIH toolbox aggression. The ASR rule-breaking scale contains an item that assesses whether a subject "gets drunk," but this does not appear to have influenced the current analysis. First, this is only one out of 14 items on the rule-breaking scale (and 40 items in total that contributed to the Externalizing factor). Second, all four scales that formed the Externalizing factor independently correlated with AUD symptom severity (all $p < 0.001$), such that each individual aspect of Externalizing appears to be related to AUD severity. Overall the model hence supports that externalizing symptoms in general mediate the causal influence of other factors on AUD.

Research has also focused on the high coincidence of AUD and internalizing disorders[41–43]. Previous causal modeling research found a causal path from internalizing disorder to drinking behavior in AUD (mediated through drinking-to-cope)[44]. The current model contains a confounded relationship between Internalizing and Conscientiousness/Attention, indicating an inability of the model to determine the relationship between these two factors, possibly due to underlying constructs (e.g. drinking motives) that were not captured in this dataset. Therefore, it remains to be further described by future research if negative affect is a common underlying cause of internalizing and AUD symptoms[41], or if there is an independent causal influence of internalizing psychopathology on AUD symptoms. It is possible that this relationship could be better examined through a longitudinal study, as pathodevelopmental perspectives on AUD have proposed that early stages of addiction are characterized by low levels of internalizing, but later stages of addiction are characterized by increasing levels of internalizing[41].

Interestingly, our results suggest an important role of fluid cognition in AUD, but this is seldom addressed in current neurobiological models of addiction. However, executive function is central to neurobiological models of addiction[9,19,38,39]. Fluid cognition (here, measured through performance on an abbreviated form of Raven's progressive matrices)[45], or a person's ability to reason and think abstractly and flexibly, has an intuitive relationship with the concept of executive function. Authors have often considered working memory to be either indicative of executive function[46], or of fluid cognition[47], and executive function and fluid cognition are similarly impacted by brain lesions[48]. Our results generate the hypothesis that fluid cognition and Crystalized IQ, including problem-solving and abstract reasoning, lie at the beginning of a causal hierarchy eventually influencing AUD severity. This fits previous empirical evidence demonstrating that AUD is associated with deficits attributed to various fluid cognitive abilities or executive function such as working memory[49,50] and planning and goal maintenance[51], but expands on this by indicating that these factors have a causal influence on AUD symptom severity. Previous research has also demonstrated that high-level cognition predicts initiation of substance use in adolescence[52], lifetime drug use and abuse[53], and addiction treatment outcomes[54]. Our model thus adds to the growing body of empirical evidence that proposes a causal role of cognition as a primary resilience factor and potential treatment target in AUD.

Our results also provide critical empirical evidence for the role of hierarchical brain network interactions in AUD. We found a direct brain-phenotype link from frontoparietal (executive) network connectivity to fluid cognition, corroborating previous evidence of this link in healthy populations[55–59]. In addiction, frontoparietal network dysfunction has been implicated in impaired inhibitory control[19,60,61] and self-regulation[27,61]. Individuals with AUD show decreased recruitment of frontoparietal

network during social–emotional processing[62,63], decision-making[64], and cognitive control[65], as well as decreased fronto-parietal connectivity during rest[66]. Frontoparietal disengagement during social–emotional processing predicts relapse in AUD[67], and decreased frontoparietal activation during inhibitory control predicts later drinking in adolescents[68,69]. Our results indicate that causal influences of frontoparietal network connectivity on AUD are mediated through deficits in overall cognitive ability and its downstream effects on the broader Negative Affect domain. A recent systematic review[70] also showed that targeting dlPFC (part of the frontoparietal network)[23] can improve cognitive deficits in addiction, including executive functions. This is particularly relevant, as our data-driven model also indicates that neuromodulation of frontoparietal network could improve executive functioning, with downstream effects on AUD severity.

We also found direct effects of cingulo-opercular and dorsal attention network connectivity onto the frontoparietal network. The cingulo-opercular network reacts to salient stimuli regardless of positive/negative valence[71,72], while the dorsal attention network supports the external focusing of attention[73] and encodes top-down control and working memory load[74]. Individuals with AUD also show decreased cingulo-opercular activation during social–emotional processing[62,63], cognitive control[65,75,76], and decision-making[64]. Our results suggest that dysfunctional connectivity in salience and attentional networks can contribute to cognitive dysfunction in AUD, with these effects being mediated through executive network connectivity. The presented causal model hence provides direct evidence for brain-directed treatment approaches targeted at the frontoparietal network, such as cognition-enhancing therapy[61,77,78], pharmacological interventions (cognitive enhancers)[77] or neuromodulation treatment (e.g., by external devices)[70,79] or neurofeedback interventions[27].

A key result generated from our CDA is the role of ventral attention/language network connectivity as a central "hub" in the brain during resting-state. The ventral attention/language network has been characterized in many brain network partitions[80–82], and while it historically has been implicated in language processing[23], more recent research has described a role in orienting attention to unexpected events[83,84]. Ventral attention/language network connectivity caused cingulo-opercular network connectivity directly, and indirectly caused dorsal attention network connectivity (mediated through cingulo-opercular connectivity, and posterior multimodal association network connectivity). Therefore, ventral attention/language network connectivity exerts influences on frontoparietal network connectivity through multiple different pathways, and might have long-range impacts on cognition and eventually on AUD severity. The potential involvement of ventral attention/language network in AUD appears to have been scarcely investigated. This network encompasses vlPFC regions that are close to left dlPFC regions that are often targeted in neuromodulation interventions for addiction[79], and therefore neuromodulation targeted at left DLPFC might also stimulate ventral attention/language networks. Left vlPFC regions are also implicated in cognitive interventions for addiction[61]. Future analysis should examine the relationship between brain ventral attention/language networks and cognitive dysfunction in AUD, and the implications for treatment.

The analysis method used in the current manuscript is not free of limitations, and other limitations are also imposed by the nature of the dataset we used. Notably, the fact that we did not find any cycles (i.e. variable $X$ causes variable $Y$, and variable $Y$ causes variable $X$) in the current data does not mean that they do not exist. The causal discovery algorithm used in this analysis cannot discover recurring cycles in cross-sectional data, but is capable of discovering recurrent cycles when more than one time

point is measured and the cycles unfold over time. Future analysis should incorporate longitudinal data to specifically test the possibility that recurrent cycles might contribute to AUD[9]. This limitation extends to the brain network subgraph we recovered as well; the causal discovery algorithm we used cannot recover bidirectional relationships in cross-sectional data, so some brain network links that are actually bidirectional processing streams may instead be represented as the predominant causal relationship between two networks. Finally, the CDA algorithm also uses a penalized likelihood score, therefore potentially missing weak causal links present in the data; however, this practice also serves to increase the confidence in the causal relationships the algorithm does find.

An important limitation of the dataset is the extremely limited assessment of the Positive Valence domain in the HCP dataset. Current perspectives in addiction emphasize the role of Positive Valence domains[9,16,19,38,39], but the HCP dataset does not contain many measures in this domain. The data did contain a measure of Delay Discounting, which had causal influences on AUD severity, but this factor appeared to be grouped with other cognitive factors and could not be interpreted as a unique measure of Reward Valuation. The HCP dataset also contained a gambling choice fMRI task, but this task did not provide a phenotypic measure of incentive salience processing. Future analysis will need to carefully measure Positive Valence domains, in addition to the domains measured in the HCP dataset, to determine where these domains fit in an overall causal model of AUD. Another limitation of the dataset is that the cross-sectional design employed by the HCP is also unable to assess certain predictions of pathodevelopmental perspectives on addiction, such as the possibility that different causal factors are involved in early and late stages of AUD[41].

To conclude, this study is the first to conduct a machine learning search for causal influences of AUD symptoms over a wide phenotypic and neurobiological space. We found phenotypic factors related to several RDoC domains, and confirmed hypothesized influences of a Negative Valence (Negative affect > Conscientiousness/Inattention > Externalizing > AUD symptom severity) and Cognitive Systems (Fluid Cognition > Crystalized IQ > Working Memory/Language/Math > Social/Affective/Psychiatric factors > AUD symptom severity) on AUD. The model proposed a hierarchy with causal influence propagating from brain function to cognition (Fluid/Crystalized Cognition, Language/Math & Working Memory) to social (Agreeableness/Social Support) to affective/psychiatric function (Negative Affect, low Conscientiousness/Attention, Externalizing symptoms) and ultimately AUD symptoms. These results underscore (a) a strong causal link between prefrontal brain function/cognition and affective/psychiatric factors and (b) an important buffer function of social factors (Social Support, Agreeableness). Our data-driven model hence confirmed hypothesized influences of cognitive and affective factors on AUD, while underscoring that traditional addiction models need to be expanded to highlight the importance of social factors, among others. Results further demonstrated that it is possible to reduce a broad phenotypic space (100 measures) to a limited set of causal factors of AUD, which can inform future research. We argue that the presented causal model of AUD provides evidence for exploring two different kinds of treatment approaches, specifically for investigating (a) "top-down" interventions aimed at enhancing high-level cognition, including brain-directed interventions targeting the executive network and (b) "integrative" interventions that take the interplay between brain/cognitive, affective/psychiatric factors, and social factors into account. We note that we did not investigate the individual heterogeneity of the causal factors involved in this model, but only provided a static causal model of an "average"

**Table 2 Demographic characteristics of the final sample ($n = 926$).**

| Demographics | Options | Total N | AUD | Control | AUD − Control difference |
|---|---|---|---|---|---|
| Gender | M | 428 | 128 | 300 | $\chi^2 = 28.74$, $p < 0.001$ |
| | F | 498 | 76 | 498 | |
| Race | White | 700 | 166 | 534 | $\chi^2 = 19.56$, $p = 0.002$ |
| | Black/African-American | 130 | 15 | 115 | |
| | Asian/Nat. Hawaiian/Other Pacific Islander | 57 | 8 | 49 | |
| | Other | 39 | 15 | 24 | |
| Age | Mean | 28.84 | 28.65 | 28.88 | $t(924) = 0.80$, $p = 0.43$ |
| | Standard deviation | 3.69 | 3.38 | 3.74 | |
| Education | Mean | 14.98 | 14.95 | 14.99 | $t(924) = 0.31$, $p = 0.76$ |
| | Standard deviation | 1.77 | 1.75 | 1.77 | |
| Income | Mean | 5.10 | 5.00 | 5.13 | $t(924) = 0.78$, $p = 0.44$ |
| | Standard deviation | 2.13 | 2.15 | 2.13 | |
| AUD symptoms | 0 | 538 | | | |
| | 1 | 184 | | | |
| | 2 | 98 | | | |
| | 3 | 46 | | | |
| | 4 | 43 | | | |
| | 5+ | 17 | | | |
| AUD diagnosis | Yes | 204 | | | |
| | No | 722 | | | |

**Table 3 List of AUD symptoms as described in the DSM.**

| DSM-IV-TR | Symptoms | DSM-5 |
|---|---|---|
| Alcohol Abuse | 1. Recurrent use of alcohol resulting in a failure to fulfill major role obligations at work, school, or home (e.g., repeated absences or poor work performance related to alcohol use; alcohol-related absences, suspensions, or expulsions from school; neglect of children or household) | Y |
| | 2. Recurrent alcohol use in situations in which it is physically hazardous (e.g., driving an automobile or operating a machine when impaired by alcohol use). | Y |
| | 3. Recurrent alcohol-related legal problems (e.g., arrests for alcohol-related disorderly conduct). | N |
| | 4. Continued alcohol use despite having persistent or recurrent social or interpersonal problems caused or exacerbated by the effects of alcohol (e.g., arguments with spouse about consequences of intoxication) | Y |
| Alcohol Dependence | 1. Need for markedly increased amounts of alcohol to achieve intoxication or desired effect; or markedly diminished effect with continued use of the same amount of alcohol. | Y |
| | 2. The characteristic withdrawal syndrome for alcohol; or drinking (or using a closely related substance) to relieve or avoid withdrawal symptoms. | Y |
| | 3. Drinking in larger amounts or over a longer period than intended. | Y |
| | 4. Persistent desire or one or more unsuccessful efforts to cut down or control drinking | Y |
| | 5. Important social, occupational, or recreational activities given up or reduced because of drinking | Y |
| | 6. A great deal of time spent in activities necessary to obtain, to use, or to recover from the effects of drinking | Y |
| | 7. Continued drinking despite knowledge of having a persistent or recurrent physical or psychological problem that is likely to be caused or exacerbated by drinking. | Y |

individual with AUD symptoms, as a first step. We believe that this initial step of describing a comprehensive, integrated, multimodal but also reduced model (in a principled data-driven way) is crucial. We see the provided causal model as a working model, which can be further expanded (e.g. by the RDoC Positive Valence factors), explored with regard to individual heterogeneity and used in predictive modeling studies on alcohol use trajectories in active users, as well as in individuals with AUD in treatment.

## Methods

**Subjects**. We analyzed data from the final release of the WU-Minn Human Connectome Project ($n = 1206$, aged 22–35, 54% females). All subjects provided written informed consent at Washington University. The CDA included all subjects who had complete data from all modalities (phenotypic $n = 933$, resting-state fMRI $n = 1085$, final $n = 926$). Subjects with AUD comprised 22% of the included sample, and 37% of subjects with AUD were females. See Table 2 for demographic characteristics of the included sample.

**Outcome measure: AUD symptom severity**. Subjects were assessed for symptoms of alcohol abuse and dependence using the Semi-Structured Assessment for the Genetics of Alcoholism (SSAGA). Symptom count data were provided for DSM-IV-TR alcohol abuse and alcohol dependence (Table 3). Symptom counts for alcohol abuse were provided as 0, 1, or 2+, and symptom counts for alcohol dependence were provided as 0, 1, 2, or 3+ (i.e. truncated symptom counts were provided). Given the low number of subjects with the highest symptom counts, it is unlikely that including a more fine-grained symptom count would have provided much additional information. It is therefore also likely that participants mostly had mild/moderate AUD severity, and that as the sample was young adults (age 22–35), this sample likely represents an early stage of AUD.

DSM-5 re-categorized alcohol abuse and alcohol dependence into a single disorder (AUD) using the criteria of both alcohol abuse and dependence, with one symptom of alcohol abuse removed (legal problems) and one symptom added (craving). We reconstructed this change by adding alcohol abuse and dependence symptom counts for each subject. Given recent interest in dimensional rather than categorical psychiatric dysfunction, we used the AUD symptom count (severity) as our primary outcome variable.

**Behavioral and self-report measures**. The HCP dataset contains a wide array of self-report, diagnostic and behavioral measures assessing domains of cognition,

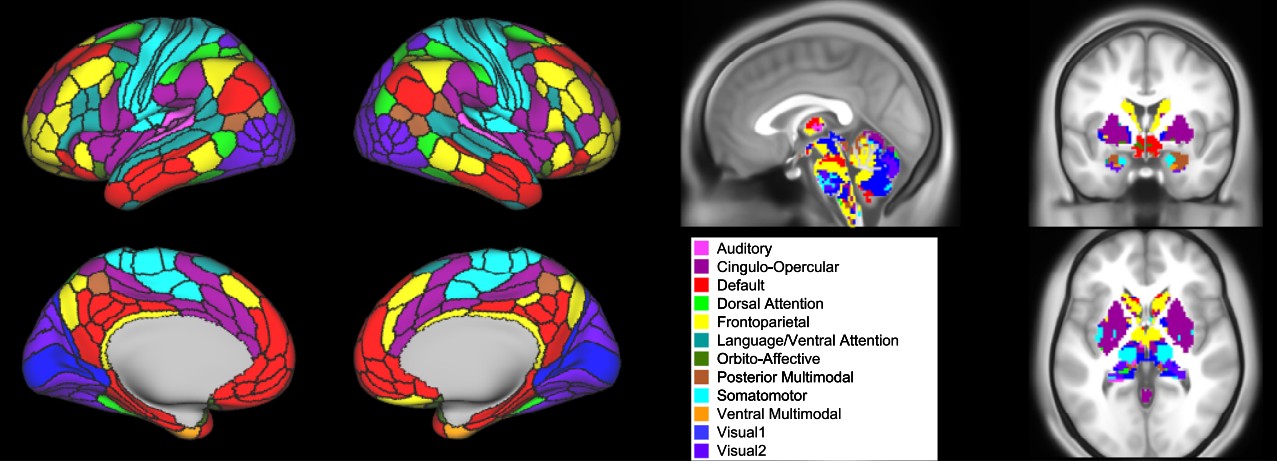

**Fig. 2 Cole-Anticevic brain-wide network partition (CAB-NP).** We conducted a whole-brain parcellation and assigned brain parcels to 12 large-scale networks according to the Cole-Anticevic brain-wide network partition (CAB-NP) parcellation[23]. This parcellation built on the Glasser multimodal cortical parcellation by including subcortical and cerebellar parcels, and assigned each of the 718 parcels to a large-scale brain network using Louvain community detection.

emotion, social function, psychiatric dysfunction, and personality. We selected all available measures from these domains for further analysis (100 in total). We included all provided measures, but when provided, we used summary scores rather than item-level or minor scores as long as the summary score encompassed the task construct of interest[85]. For example, for the Short Continuous Penn Test, we included the summary statistics of sensitivity and specificity but not more specific scores, and for delay discounting we included the area under the curve for $200 and $40k but not the individual discounting levels. For behavioral tasks completed in the scanner, we included separate behavioral measures for each of the major conditions. Note that in the interest of more completely characterizing the phenotypic space measured in the HCP data, we included behavior from tasks completed in the scanner but not fMRI measures of task-related brain activation.

If both age/gender-adjusted and unadjusted scores were provided, we included only the adjusted scores. For example, data from the Achenbach self-report inventory included raw scale scores, and scale scores adjusted for age and gender; given the strong relationship between psychiatric symptomology and age/gender, we included the age/gender-adjusted measures from this self-report inventory. This age/gender adjustment was thus applied to 14/100 (14%) of variables. Data from the NIH Toolbox cognition battery were likewise adjusted for participant age, given the strong dependency between age and cognitive abilities. Finally, the sensory sensitivity battery provided by the HCP consortium included measures of noise, taste, smell, and pain intensity, likewise controlled for the influence of age. Thus, age adjustment was applied to 11/100 (11%) of variables. Therefore in sum, 25% of variables were adjusted in some manner for participant characteristics (age: 11%, age/gender: 14%), while the remaining 75% of variables were not adjusted for any participant characteristics. We also excluded all items that were linear combinations of other data. For a list of included variables and excluded variables see Supplementary Data 6; for descriptive statistics of the included variables see Supplementary Data 7.

**Factor analysis of phenotypic data.** To reduce the phenotypic space measured in the HCP to a set of underlying domains, we conducted an EFA in the entire HCP sample that had complete phenotypic data ($n = 933$, 53.5% females). Factors were extracted using maximum likelihood as calculated by the expectation-maximization algorithm, and we used an oblimin rotation to allow for correlated factors. EFA was calculated in R using the "psych" package[86]. The choice of EFA over similar data reduction schemes such as PCA was made because EFA explicitly accounts for error due to unreliability in measurement[87] unlike PCA[88]. Furthermore, maximum likelihood extraction is robust against violation of distributional assumptions [as discussed in refs. [89,90]], providing a powerful technique for reduction of questionnaire and other behavioral data. As factor analysis conducts an eigenvalue decomposition of the correlation matrix among items, this analysis also does not assume that items submitted to factor analysis are measured on similar scales. Oblimin rotation allows for correlated factors, which is critical to data reduction over a large phenotypic variable space as we expect many factors to be closely related but separable (for example, negative affect and internalizing psychopathology). We used Monte Carlo permutation analysis (parallel analysis)[91] to determine how many factors were statistically significant at $p < .05$[92]. Monte Carlo simulation was also calculated using the "psych" package for R[86]. While some methods for choosing the number of factors in a solution are not robust against violation of distributional assumptions, the use of permutation analysis to choose the number of

extracted factors ensures that this selection does not depend on any assumptions regarding underlying distributional qualities.

**Resting-state fMRI acquisition and preprocessing.** High-resolution structural and functional MRI data were collected on a Siemens 3T Connectome Skyra scanner with a 32-channel head coil at Washington University. See Uğurbil et al.[21] for a full description of the acquisition parameters for rfMRI in the HCP database. Resting-state fMRI (rfMRI) was collected over 2 days in four runs of 14:33 each. Structural data were preprocessed as described in Glasser et al.[93], using the most recent version of the HCP preprocessing pipeline (4.1). Briefly, anatomical image preprocessing consisted of bias field and gradient distortion correction, coregistration of T1w and T2w images, and linear and nonlinear registration to MNI space. Cortical surfaces were constructed using FreeSurfer. Surface files were transformed to MNI space, registered to the individual's native-mesh surfaces, and downsampled.

Functional MRI preprocessing is fully described in Glasser et al.[93]. Briefly, volumetric fMRI were subjected to gradient distortion correction, motion correction, and referencing to T1w. All transforms were concatenated and run in a single nonlinear resampling to MNI space. Data were then masked by the PostFreeSurfer brain mask and normalized. This volumetric timeseries was then mapped to a combined cortical surface and subcortical voxel space ("grayordinates") and smoothed with a 2 mm FWHM Gaussian.

Finally, fMRI data were high-pass filtered (FWHM = 2355 s) and cleaned of artifacts using ICA-FIX[94,95]. Artifact components and 24 motion regressors[96] were regressed out of the data in a single step. This produced the final ICA-FIX denoised versions of the data in both volumetric and CIFTI (combined grayordinates and subcortical/cerebellar voxels) space (https://www.humanconnectome.org/study/hcp-young-adult/document/1200-subjects-data-release). CIFTI data were used for all primary analyses, while a replication analysis used the volumetric data to ensure the results replicated with multiple different brain network measures.

**rfMRI parcellation and network assignment.** Our analysis of rfMRI network connectivity only included subjects who completed at least one full day of rfMRI acquisition (two runs of 14:33), and 94% of subjects had four full runs of rfMRI data. We parcellated the whole brain, including cortex, subcortex, and cerebellum, into 718 parcels using the Cole-Anticevic brain-wide network partition (CAB-NP)[23], a parcellation scheme that builds on the Glasser et al.[97] multimodal cortex parcellation (360 parcels). We chose the CAB-NP parcellation because while the Glasser parcellation used multiple measures including myelination, rfMRI activity, and anatomical landmarks to delineate a fine-grained map of cortical space, it did not include any subcortical voxels, and did not explicitly assign parcels to large-scale networks using principled statistical methods. The CAB-NP parcellation built on the Glasser parcellation by (1) assigning subcortical and cerebellar voxels to parcels and (2) by using Louvain community detection to delineate 12 large-scale networks consisting of cortical, subcortical, and cerebellar regions (Fig. 2).

As the choice of parcellation can influence network connectivity profiles[98], we furthermore investigated whether the results of this part of the analysis would replicate when using an independently derived set of resting-state networks. We used the set of 300 regions-of-interest (ROIs) and associated network assignments described in ref. [82], which included the 264 ROIs described in ref. [81] with added subcortical and cerebellar ROIs (Supplementary Fig. S4). This parcellation used a

**Fig. 3 Discovery of causal orientations using conditional independence relationships.** Four different ways that three variables X, Y, and Z could be causally related. **a** is a structure known as a "collider," in which X and Y both cause Z, but X and Y are not related. In this structure, X and Z are dependent, and Y and Z are dependent, while X and Y are independent. However, when Z is conditioned on (controlled for), X and Y are dependent. Meanwhile, in panel **b**, Z causes both X and Y. In this structure, X and Y are dependent because of their common cause, and are independent when Z is conditioned on. In panels **c, d**, X and Y are dependent because one causes Z, which then causes the other. In both of these panels, X and Y are independent when Z is conditioned on, as Z is the only link from X to Y. GFCI utilizes conditional independence tests to determine causal direction in graph edges, specifically by identifying "collider" cases in the graph (since these cases imply different conditional dependencies than the other three cases).

set of functionally defined spherical ROIs in volumetric space, and employed the Infomap algorithm to compute network assignments for each ROI. As this analysis was intended as a replication of our primary results, we only selected the networks that aligned closely with the CAB-NP networks (Supplementary Table S2).

**Calculation of rfMRI network connectivity**. For each network, we computed pairwise Pearson correlations between each pair of parcels in the network. Pearson correlations were transformed to approximate a normal distribution using Fisher's z-transform. Within each of the networks, we then took the average of the parcel-to-parcel correlations to obtain a summary statistic for within-network connectivity[99–103]. This procedure therefore summarizes how tightly connected (coherent) the regions comprising each of the 12 networks are with each other. This resulted in a set of average network-level connectivity values for each subject.

**CDA: Greedy Fast Causal Inference**. Causal models represent, often graphically, the set of cause-and-effect relationships that are present within a set of data[104]. As the number of variables in a dataset increases, so too does the space of possible causal models that could give rise to the observed data, making the problem of identifying which of the potential causal models best fits the observed data very difficult. CDA uses machine learning to determine which causal models are best supported by the data[105]. There are many CDA algorithms that make a wide variety of assumptions and have varying performance characteristics; for review, see Glymour et al.[106].

In the current study, we applied GFCI[24], an accurate and fast algorithm for establishing causal relationships from data even in the presence of unmeasured confounds. GFCI operates in two phases. GFCI begins by searching the space of possible graphs to create a preliminary graph that minimizes a penalized likelihood score, in this case the Bayesian Information Criteria[107]. This initial search phase is done using the Fast Greedy Equivalence Search method[108]. After the initial search phase, the algorithm refines the discovered graph by conducting a series of conditional independence tests. This phase rules out any edges that imply conditional dependencies not borne out by the data (for an example, see Fig. 3). Specifically, this phase capitalizes on the fact that "collider" structures (Fig. 3a; a case where two variables both cause a third variable) imply a separate set of conditional dependencies than any other ways those three variables could be causally related (Fig. 3b–d). The most important distinction of GFCI compared to other causal discovery methods is that GFCI can detect confounding factors, and as such is particularly suited to analysis of real-world data, where there is no guarantee that every relevant variable has been measured. The output of the GFCI algorithm is a partial ancestral graph with edge types indicating causal relationships, uncertain relationships, and the presence of unmeasured confounding variables. GFCI analysis was implemented using Tetrad. Analysis was run with default parameters; that is, using alpha = 0.01, maximum degree of the graph = 100, and a penalty discount of two. Penalized likelihoods for models were calculated using the Bayesian Information Criteria[107], which is the default model fit index in Tetrad and the most commonly used model fit index in CDA.

To recover effect sizes of the edges in the model, we fit a structural equation model (SEM) to the graph structure using the "lavaan" package for R[109]. We present the graph GFCI learned from the full dataset with SEM effect sizes for each edge. As an additional analysis of model stability in smaller samples, we also conducted a stability analysis by resampling 90% of the sample[110] without replacement with 1000 repetitions (jackknife). Finally, to test the replicability of the brain model, we used the R package "lavaan" to fit a SEM representing the discovered causal model (CAB-NP)[23] to the alternately derived brain network data (Greene-300)[82].

**Statistics and reproducibility**. Factor analysis (parallel analysis and factor extraction) was done using the "psych" package (version 2.0.12)[86], and factor rotation used the "GPArotation" package (version 2014.11-1) for R (version 3.6.3). FMRI network connectivity analysis was done using Connectome Workbench

(version 1.4.2) and MATLAB (version R2018b). Causal discovery analysis (GFCI) was done using Tetrad (version 6.7). Stability analysis (jackknifing) was done in Tetrad (version 6.7) using 90% subsampling and 1000 replications. Effect size calculation was done using the "lavaan" package (version 0.6-6) running in R (version 3.6.3). We used a very large, well-documented, and publicly available dataset (HCP; $n = 926$) to ensure the representativeness of the analysis, and we additionally validated the stability of the neurobiological analyses using a secondary, independently derived brain parcellation, and the stability of the causal discovery analyses using a jackknifing procedure.

**Reporting summary**. Further information on research design is available in the Nature Research Reporting Summary linked to this article.

## Data availability
The data analyzed for this study are available from the WU-Minn Human Connectome Project Consortium (https://www.humanconnectome.org/study/hcp-young-adult/document/1200-subjects-data-release). These data are available for public use with a data use agreement. Raw data sufficient to reproduce the GFCI and SEM results is available (Supplementary Data 8).

## Code availability
All code and analysis packages used in this manuscript are freely available with the exception of MATLAB. MATLAB was used for convenience (fMRI network correlations) but is not required to replicate analyses. Connectome Workbench is freely available from https://www.humanconnectome.org/software/get-connectome-workbench. R is freely available from https://www.r-project.org/. The R packages "psych", "GPArotation", and "lavaan" can be freely installed within R. Tetrad is freely available from https://www.ccd.pitt.edu/tools/.

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

## Acknowledgements

E.R. is supported by National Institutes of Mental Health (NIMH) grant T32-MH115866. E.K. received support for this work from the National Center for Advancing Transla-tional Sciences of the National Institutes of Health Award Number UL1TR000114. A.Z. is supported by the P30 DA048742-01A1 of the National Institute on Drug Abuse. The content is solely the responsibility of the authors and does not necessarily represent the official views of the National Institutes of Health or the National Institutes of Mental Health. The authors thank Matthew Kushner for valuable input and editorial assistance.

## Author contributions

E.R.: conceptualization, methodology, software, validation, formal analysis, investigation, writing—original draft, writing—review & editing, visualization. E.K.: conceptualization, methodology, software, validation, writing—review & editing, visualization, supervision. A.Z.: conceptualization, methodology, validation, resources, writing—review and editing, supervision, project administration.

## Competing interests

The authors declare no competing interests.
