## [Peer Review File · Communications Biology]

Reviewers' comments:

Reviewer #1 (Remarks to the Author):

The study aims to the identification of a neurobehavioural model that explains Alcohol Use Disorder. The authors apply the Causal Discovery Analysis approach to analyse behavioural and neuroimaging data from a large cohort of AUD subjects, identifying a hierarchical model of causal influence that encompasses brain connectivity networks to cognitive domains to social to psychiatric functions resulting in AUD symptoms.

The implemented approach is novel, and the study's findings would be an interesting addition to the literature. However, several points need addressing.

1. The authors select a comprehensive array of diagnostic and behavioural measures corresponding to AUD subjects' sample, favouring a complete characterisation of the AUD profile. However, given the large number of psychological assessments with different score scales and that these will be later analysed by exploratory factor analysis, I wonder whether the authors implemented a statistical manipulation of each assessment's scores, allowing the comparison of data with different distributions.

2. The manuscript specifies that when adjusted and unadjusted data was available, the authors considered only adjusted data. I would ask the authors to specify whether some data reported the unadjusted score only when the assessment required an adjusted score. Furthermore, I would ask the authors to specify the percentage of studies that provided adjusted data only, unadjusted data only, and adjusted + unadjusted data.

3. I am puzzled by the authors' decision to specify in table 1 the population race, as the relationship between race and AUD is not discussed in the manuscript, or the race variable is included in any steps of the study analyses. I think that specifying the sample's mean education would be more informative, as the behavioural measures included tests adjusted for age and education.

4. Providing the list of AUD symptoms selected for the study in the main text could improve the manuscript's clarity and readability.

5. The analysis implemented by the authors is appropriate and highly informative on the connectivity and phenotypic causal influences to AUD symptoms. However, the discussion of the CDA findings may appear contradictory. For instance, the CDA analysis reveals a hierarchical model where the brain connectivity networks and the behavioural phenotypes separate into two clusters interconnected via the link between the frontoparietal network and the Fluid Cognition domain. In the brain connectivity network cluster, the Language network exerts influences on the Frontoparietal network. In contrast, in the behavioural phenotypes cluster, the Languages tasks seem to be caused by Fluid Cognition, Visuospatial, and Crystallized IQ, which are associated with the Frontoparietal network.

I would ask the authors a clarification on the interpretation of the CDA results. As on page 6, line 133, the manuscript mention to "in-scanner" tasks, does phenotypic data refer to behavioural measures only or behavioural tasks-related functional activations as well?

I wondered whether the authors considered using the functional maps of each phenotypic domain in the Causal Discovery Analysis rather than the posterior parcellation of resting-state data, which might help the interpretation of the CDA findings.

Reviewer #2 (Remarks to the Author):

The study of Rawls et al adopts a data-driven approach to elucidate the neural correlates of alcohol use disorder. The authors use a comprehensive imaging data set and behavioral measurements to undertake this task. The authors use Causal Discovery Analysis and uncover a hierarchy of causal

influences from cognition to social factors to affective function to alcohol use severity. The manuscript is well written and the results are well presented. However, there might be some additional visualization that may enhance the interpretability of the results and inform the reader about their stability. Moreover, it might be worthwhile to check to what extent the choice of the brain parcellation influences the results. It is reported in the literature that such choice can have a rather substantial effect.

Concerning the visualization of the stability of the results: It would be very useful if the authors rendered a visualization as the one in Figure 3, but with the uncertainty of the SEM edge weights visualized. For instance, the edges could have different thickness that is inversely related to the stability of the measurements. The authors report results of a jackknife procedure but in the form of a table. Tables are in general hard to parse and use as an overview. Therefore, a visualization of the uncertainty of the causal relations in the uncovered hierarchy of causal influences would be very informative.

Concerning the parcellation choice: The authors provide meaningful justification for the use of the Cole-Antecic parcellation. However, it is known that the choice of the parcellation can significantly affect connectivity results (e.g., Messe 2019 Human Brain Mapping). It would be, thus, advisable to demonstrate that the main conclusions do not depend on the choice of this particular parcellation scheme. For instance, the authors could re-run the analysis with the Lewis et al 2019 parcellation (<https://doi.org/10.1101/2019.12.20.883553>). This parcellation is just an example. The authors could use any other alternative that they find of use.

I hope that the above remarks are of help.

We would like to thank the two reviewers for what are clearly a well thought out and carefully considered set of comments based on a thorough read of our manuscript. In the following response to reviewers, the revised manuscript, and the revised supplement, you will find that we have undertaken considerable efforts to respond fully and comprehensively to each concern raised by reviewers. We believe the new paper is more methodologically sound and greatly improved in no small part due to the helpful comments and suggestions from both reviewers.

The following is a summary of the additional information we have provided and additional analyses we ran to meet reviewer expectations. First, we have taken particular care to respond to Reviewer 1's comments on the use of exploratory factor analysis across data from multiple different scales. We have run additional analyses and verified that putting these data on a similar scale (z-scores) does not change the results of the factor solution. We also added additional information as requested by Reviewer 1 on the demographics of the sample and the method of adjusting data used by HCP. We also thank Reviewer 2 for insightful feedback regarding visualization of stability analyses, and the potential influence of brain parcellation on our reported results. We have taken care to ensure that the results reported in our manuscript replicate across an independently-derived brain network parcellation, and we have prepared additional visualizations for this alternate parcellation analysis and regarding the stability analyses of the causal discovery results.

Find below a point-by-point response to the concerns raised by each reviewer. Throughout, we format our comments in blue to separate them from the reviewer comments, and include all modified text and figures for the reviewer's benefit.

Reviewer #1 (Remarks to the Author):

The study aims to the identification of a neurobehavioural model that explains Alcohol Use Disorder. The authors apply the Causal Discovery Analysis approach to analyse behavioural and neuroimaging data from a large cohort of AUD subjects, identifying a hierarchical model of causal influence that encompasses brain connectivity networks to cognitive domains to social to psychiatric functions resulting in AUD symptoms. The implemented approach is novel, and the study's findings would be an interesting addition to the literature. However, several points need addressing.

1. The authors select a comprehensive array of diagnostic and behavioural measures corresponding to AUD subjects' sample, favouring a complete characterisation of the AUD profile. However, given the large number of psychological assessments with different score scales and that these will be later analysed by exploratory factor analysis, I wonder whether the authors implemented a statistical manipulation of each assessment's scores, allowing the comparison of data with different distributions.

The reviewer makes an excellent point, that submitting data to a unified analysis framework such as exploratory factor analysis might suggest that data should be measured in some common scale. Fortunately, EFA makes no assumption of such a common measurement scale, and is therefore a powerful method for analysis of data on multiple scales such as the presented work. As EFA decomposes the correlation matrix of individual measures, rather than relying on raw (unscaled) scores themselves, EFA does not depend on different scaling across items. To show evidence for this claim, we repeated our factor analysis using the same methods presented in the main manuscript, but submitting z-scored individual items to the same analysis, thus ensuring that each item is measured on the same scale prior to factor analysis (i.e. every item has a mean of 0 and a standard deviation of 1). This analysis indicated that the factor results remained entirely unchanged, as expected. We provide the results of this additional factor analysis in full, including a new parallel analysis and factor loadings matrix, for the reviewer's benefit. We hope this additional validation of our exploratory factor analysis will serve to assure the reviewer that our data reduction results do not depend on different scaling of items submitted to the factor analysis. We have included a figure demonstrating the results of the parallel analysis on z-scored variables below;

the new loadings table is provided as a supplemental file (New_Factor_Loadings_Zscore.xlsx) due to the size of the table.

Supplemental figure for review. Demonstration that different scaling of items submitted to factor analysis does not influence the results of EFA.

Furthermore, we explicitly chose factor analysis using maximum likelihood extraction via expectation-maximization (and permutation-based selection of the number of factors) because these methods are highly robust against what might otherwise be violations of distributional assumptions. We have added a discussion of these qualities to the test. The added discussion is included below for the reviewer’s benefit:

“Furthermore, maximum likelihood extraction is robust against violation of distributional assumptions [as discussed in (30,31)], providing a powerful technique for reduction of questionnaire and other behavioral data. As factor analysis conducts an eigenvalue decomposition of the correlation matrix among items, this analysis also does not assume that items submitted to factor analysis are measured on similar scales.” LINES 161-165

“While some methods for choosing the number of factors in a solution are not robust against violation of distributional assumptions, the use of permutation analysis to choose the number of extracted factors ensures that this selection does not depend on any assumptions regarding underlying distributional qualities.” LINES 170-173

2. The manuscript specifies that when adjusted and unadjusted data was available, the authors considered only adjusted data. I would ask the authors to specify whether some data reported the unadjusted score only when the assessment required an adjusted score. Furthermore, I would ask the authors to specify the percentage of studies that provided adjusted data only, unadjusted data only, and adjusted + unadjusted data.

The reviewer brings up an extremely important concern regarding adjustment of data as provided by the HCP consortium. The HCP consortium provided adjusted scores for all scales with a standardized adjustment; that is, there were no scores missing such standardization when it is required. For example, all psychiatric scales (Achenbach self-report) were scaled to t-scores by controlling for gender and age effects, given strong influences of gender and age in psychiatric distress scales. The cognition measures from the NIH toolbox were also controlled for age effects, given known effects of age on cognitive ability. No other measures were adjusted for any participant characteristics. We recognize that this additional information regarding adjustment of data is of benefit to the reader, and as such we have added information regarding the percentage of data that is adjusted for participant characteristics, as well as which characteristics it is adjusted for. For the reviewer’s benefit, the added text from this edit is as follows:

“For example, data from the Achenbach self-report inventory included raw scale scores, and scale scores adjusted for age and gender; given the strong relationship between psychiatric symptomology and age/gender, we included the age/gender adjusted measures from this self-report inventory. This age/gender adjustment was thus applied to 14/100 (14%) of variables. Data from the NIH Toolbox cognition battery was likewise adjusted for participant age, given the strong dependency between age and cognitive abilities. Finally, the sensory sensitivity battery provided by the HCP consortium included measures of noise, taste, smell, and pain intensity, likewise controlled for the influence of age. Thus, age adjustment was applied to 11/100 (11%) of variables. Therefore in sum, 25% of variables were adjusted in some manner for participant characteristics (age: 11%, age/gender: 14%), while the remaining 75% of variables were not adjusted for any participant characteristics.” LINES 142-151

3. I am puzzled by the authors’ decision to specify in table 1 the population race, as the relationship between race and AUD is not discussed in the manuscript, or the race variable is included in any steps of the study analyses. I think that specifying the sample’s mean education would be more informative, as the behavioural measures included tests adjusted for age and education.

We agree with the reviewer that inclusion of additional demographics information would be useful, and is indeed standard in addiction studies. As such we added 1) education and 2) income level to our demographics table (LINE 108). At this point, it is important to note that none of the measures in the study were adjusted for education; the only demographic variables considered for adjustment were gender and age.

However, we opted to continue including racial makeup of our sample in the demographics table. While race is not included in any of our analyses, it is generally recommended that authors report the racial demographics of their sample in order to assess whether the sample is representative of the population of interest. This representativeness was an aim of the Human Connectome Project as a whole, and in the authors experience racial demographics are generally included in papers reporting psychiatric outcomes of interest (such as AUD severity in our analysis). While participant race does not factor into any of our hypotheses or play a part in any of our statistical tests, it is nevertheless possible that the racial makeup of the sample would play a part in reproducibility of our findings, and thus we hope that this reporting will be helpful to other researchers.

4. Providing the list of AUD symptoms selected for the study in the main text could improve the manuscript’s clarity and readability.

We agree that a presentation of clinical symptoms of AUD would greatly aid interpretation of this manuscript; this list is now presented in the main manuscript in Table form (Table 2). For the reviewer’s benefit, find the new table below as well:

Table 2. List of AUD symptoms as described in the DSM.

DSM-IV-TR	Symptom	DSM5
Alcohol Abuse	1. Recurrent use of alcohol resulting in a failure to fulfill major role obligations at work, school, or home (e.g., repeated absences or poor work performance related to alcohol use; alcohol-related absences, suspensions, or expulsions from school; neglect of children or household).	Y
	2. Recurrent alcohol use in situations in which it is physically hazardous (e.g., driving an automobile or operating a machine when impaired by alcohol use).	Y
	3. Recurrent alcohol-related legal problems (e.g., arrests for alcohol-related disorderly conduct).	N
	4. Continued alcohol use despite having persistent or recurrent social or interpersonal problems caused or exacerbated by the effects of alcohol (e.g., arguments with spouse about consequences of intoxication).	Y
Alcohol Dependence	1. Need for markedly increased amounts of alcohol to achieve intoxication or desired effect; or markedly diminished effect with continued use of the same amount of alcohol.	Y

2. The characteristic withdrawal syndrome for alcohol; or drinking (or using a closely related substance) to relieve or avoid withdrawal symptoms.	Y
3. Drinking in larger amounts or over a longer period than intended.	Y
4. Persistent desire or one or more unsuccessful efforts to cut down or control drinking	Y
5. Important social, occupational, or recreational activities given up or reduced because of drinking.	Y
6. A great deal of time spent in activities necessary to obtain, to use, or to recover from the effects of drinking.	Y
7. Continued drinking despite knowledge of having a persistent or recurrent physical or psychological problem that is likely to be caused or exacerbated by drinking.	Y

LINE 126

5. The analysis implemented by the authors is appropriate and highly informative on the connectivity and phenotypic causal influences to AUD symptoms. However, the discussion of the CDA findings may appear contradictory. For instance, the CDA analysis reveals a hierarchical model where the brain connectivity networks and the behavioural phenotypes separate into two clusters interconnected via the link between the frontoparietal network and the Fluid Cognition domain. In the brain connectivity network cluster, the Language network exerts influences on the Frontoparietal network. In contrast, in the behavioural phenotypes cluster, the Languages tasks seem to be caused by Fluid Cognition, Visuospatial, and Crystallized IQ, which are associated with the Frontoparietal network. I would ask the authors a clarification on the interpretation of the CDA results.

The authors recognize that the use of the naming convention “language network” caused confusion in the initial version of this manuscript. We have added significant information to the discussion of the results, pointing out as our primary reasoning that the “language network” as described in Ji et al. (2019) is the direct correlate of the commonly described “ventral attention network” from prior publications. Furthermore, in a demonstration that results replicate with a separate parcellation scheme, we used an alternate parcellation that defined the ventral attention network rather than the “language” network, with equivalent model fit results. Throughout the text, we have replaced all reference to the “language network” with “ventral attention/language network” to make this clear to readers. For the reviewer’s benefit we include the modified paragraph below (changes highlighted in yellow):

“A key result generated from our causal discovery analysis is the role of ventral attention/language network connectivity as a central “hub” in the brain during resting-state. The ventral attention/language network has been characterized in many brain network partitions (40,41,109), and while it historically has been implicated in language processing (24), more recent research has described a role in orienting attention to unexpected events (110,111). Ventral attention/language network connectivity caused cingulo-opercular network connectivity directly, and indirectly caused dorsal attention network connectivity (mediated through cingulo-opercular connectivity, and posterior multi-modal association network connectivity). Therefore, ventral attention/language network connectivity exerts influences on frontoparietal network connectivity through multiple different pathways, and might have long-range impacts on cognition and eventually on AUD severity. The potential involvement of ventral attention/language network in AUD appears to have been scarcely investigated. This network encompasses vIPFC regions that are close to left dlPFC regions that are often targeted in neuromodulation interventions for addiction (108), and therefore neuromodulation targeted at left DLPFC might also stimulate ventral attention/language networks. Left vIPFC regions are also implicated in cognitive interventions for addiction (90). Future analysis should examine the relationship between brain ventral attention/language networks and cognitive dysfunction in AUD, and the implications for treatment.” LINES 492-507

6. As on page 6, line 133, the manuscript mention to “in-scanner” tasks, does phenotypic data refer to behavioural measures only or behavioural tasks-related functional activations as well?

We appreciate the reviewer’s attention to our potentially confusing wording in this statement. We changed the wording of this sentence to be clearer that we used behavioral measures from tasks that subjects completed in the scanner but not task-related fMRI activations. The changed text is as follows:

“For behavioral tasks completed in the scanner, we included separate behavioral measures for each of the major conditions. Note that in the interest of more completely characterizing the phenotypic space measured in the HCP data, we included behavior from tasks completed in the scanner but not fMRI measures of task-related brain activation.” LINES 137-140

7. I wondered whether the authors considered using the functional maps of each phenotypic domain in the Causal Discovery Analysis rather than the posterior parcellation of resting-state data, which might help the interpretation of the CDA findings.

While it is possible that interesting information could be potentially gleaned from the inclusion of task-fMRI data in a causal model, it is the authors’ belief that such inclusion is beyond the scope of the current manuscript. The HCP consortium collected data from several fMRI tasks in-scanner, but these tasks represent only a small portion of the wide phenotypic space of variables assessed by the HCP consortium. The most important factors in our model, such as fluid intelligence, negative affect, externalizing, and social function, were not assessed by any of the included tasks. Furthermore, our approach based on brain network connectivity is ill-suited for inclusion of task-fMRI maps, as the interpretation of within-network connectivity might be less clear for task-related fMRI than for resting fMRI. We believe that our response to concern 5 (above) will serve to make the interpretation of the causal modeling more clear, as the justifiable confusion regarding the naming of the “language” network appears to have caused a bulk of the difficulty interpreting the causal model results.

Reviewer #2 (Remarks to the Author):

The study of Rawls et al adopts a data-driven approach to elucidate the neural correlates of alcohol use disorder. The authors use a comprehensive imaging data set and behavioral measurements to undertake this task. The authors use Causal Discovery Analysis and uncover a hierarchy of causal influences from cognition to social factors to affective function to alcohol use severity. The manuscript is well written and the results are well presented. However, there might be some additional visualization that may enhance the interpretability of the results and inform the reader about their stability. Moreover, it might be worthwhile to check to what extent the choice of the brain parcellation influences the results. It is reported in the literature that such choice can have a rather substantial effect.

1. Concerning the visualization of the stability of the results: It would be very useful if the authors rendered a visualization as the one in Figure 3, but with the uncertainty of the SEM edge weights visualized. For instance, the edges could have different thickness that is inversely related to the stability of the measurements. The authors report results of a jackknife procedure but in the form of a table. Tables are in general hard to parse and use as an overview. Therefore, a visualization of the uncertainty of the causal relations in the uncovered hierarchy of causal influences would be very informative.

The reviewer correctly suggests that a graphical visualization of the stability results would be informative, as opposed to presenting the stability analysis in table format only. To meet this expectation, we made a visualization analogous to Figure 3 but replaced the effect size with the stability of the jackknifed analysis. We added additional metrics to more thoroughly quantify the stability of our results as well. Specifically, for each edge we now report two stability metrics: the first is the stability of the edge existing at all (regardless of direction), and the second is the stability of the edge’s direction, as displayed in the discovered graph. We make this improved figure, and a corroborating table of stability results, available in the supplement for the manuscript.

Additionally, we present the modified table and the new figure below for the reviewer's benefit:

Figure S3. Graphical results of jackknife stability analysis. Each directed edge receives two indices of stability: the first is an indication of how often the edge is present at all in jackknifed repetitions (regardless of orientation; 1 being 100% of repetitions), and the second number (in parentheses) is an indication of how often the edge is present and correctly oriented in jackknife repetitions (unoriented edges receive only one index of stability). Note that all edges were significant ($p < .001$) in the SEM; therefore, this analysis does not characterize significance of edges but rather only provides an additional metric of stability under resampling. Notably, the majority of edges in the graph existed in a very high percentage of resampled analyses. Somewhat lower stability was found for edges between (as opposed to within) different domains, as expected based on the correlation-based grouping of domains. Interestingly, a small number of edges that were extremely stable in existence had relatively lower directional stability; this suggests (but does not demonstrate) a potential reciprocal relationship between variables that could potentially be elucidated in longitudinal data. For example, the edge between social support and negative affect existed in 100% of resampling analyses, but in many resamples this edge instead went from negative affect to social support, indicating a potential bidirectional relationship.

Table S9. Stability of causal model edges under resampling (1000 repetitions, 90% resample). Each directed edge receives two indices of stability: the first is an indication of how often the edge is present at all in jackknifed repetitions (regardless of orientation; 1 being 100% of repetitions), and the second number (in parentheses) is an indication of how often the edge is present and correctly oriented in jackknife repetitions.

Var1	Var2	Existence	Direction
PosteriorMM	Visual1	1	0.445
Visual1	VentralMM	0.741	0.741
Language	VentralMM	0.997	0.994
Visual1	Auditory	1	0.872
Auditory	Somatomotor	1	0.931
Language	PosteriorMM	1	0.512
VentralMM	Orbitoffective	0.999	0.999
CinguloOpercular	Orbitoffective	1	1
Somatomotor	Visual2	1	1
DorsalAttention	Visual2	1	0.844
Language	DefaultMode	1	0.99
Frontoparietal	DefaultMode	1	1
Language	CinguloOpercular	0.982	0.31
PosteriorMM	DorsalAttention	0.998	0.941
CinguloOpercular	DorsalAttention	0.992	0.792
CinguloOpercular	Frontoparietal	0.998	0.714
DorsalAttention	Frontoparietal	1	0.848
Frontoparietal	FluidCognition	0.58	0.49
FluidCognition	Visuospatial	0.998	0.766
FluidCognition	CrystallizedIQ	1	0.765
Visuospatial	ProcessingSpeed	0.948	0.907
MemoryPerformance	ProcessingSpeed	1	0.933
Visuospatial	LanguageTask	0.992	0.778
CrystallizedIQ	LanguageTask	1	0.765
CrystallizedIQ	DelayDiscounting	0.997	0.962
ProcessingSpeed	RelationalTask	1	0.939
LanguageTask	MemoryPerformance	1	0.854
LanguageTask	Agreeableness	0.558	0.528
DelayDiscounting	Agreeableness	0.421	0.337
RelationalTask	GambleRT	0.542	0.407
MemoryPerformance	GambleRT	1	0.958
MemoryPerformance	SocialSupport	0.581	0.488
Agreeableness	SocialSupport	0.547	0.33
Agreeableness	Externalizing	0.668	0.573
Conscientiousness	Externalizing	1	0.523
SocialSupport	Avoidance	1	0.998
Internalizing	Avoidance	1	0.992
SocialSupport	NegativeAffect	1	0.519
SocialSupport	PositiveAffect	1	0.989
Externalizing	PositiveAffect	0.988	0.988

NegativeAffect	Internalizing	1	0.519
NegativeAffect	Conscientiousness	0.285	0.25
Internalizing	Somaticism	1	0.409
Conscientiousness	Somaticism	0.94	0.483
Externalizing	AUD	1	0.881
Language	Somatomotor	0.649	*
CinguloOpercular	Auditory	0.747	*
Language	Auditory	1	*
CrystallizedIQ	Visuospatial	1	*
Internalizing	Conscientiousness	1	*

SUPPLEMENT LINES 62-83

2. Concerning the parcellation choice: The authors provide meaningful justification for the use of the Cole-Anticevic parcellation. However, it is known that the choice of the parcellation can significantly affect connectivity results (e.g., Messe 2019 Human Brain Mapping). It would be, thus, advisable to demonstrate that the main conclusions do not depend on the choice of this particular parcellation scheme. For instance, the authors could re-run the analysis with the Lewis et al 2019 parcellation (<https://doi.org/10.1101/2019.12.20.883553>). This parcellation is just an example. The authors could use any other alternative that they find of use.

The reviewer has correctly and helpfully identified that it is important to see that the results described here replicate when using an independent brain parcellation. As such, we used a recent alternative brain network parcellation (Greene-300) to demonstrate the robustness of the causal model we describe in this manuscript. The Greene-300 brain network parcellation used a set of spherical ROIs in volume space (as compared to Cole-Anticevic parcellation, which used CIFTI grayordinates), and networks were assigned using a different algorithm (Infomap). We matched the Greene-300 networks to the Cole-Anticevic networks (as described in Supplement Table S9), to show that our discovered causal model replicates extremely well in this new parcellation by fitting a structural equation model (SEM) representing the original causal model to data derived from the new parcellation. We found that the SEM had good fit metrics, despite the fact that the causal model was learned from a different brain parcellation. Furthermore, every edge as discovered in the Cole-Anticevic parcellation was highly significant ($p < .001$) in the SEM fit to the alternate parcellation. As such, we provide strong evidence that our results are robust across multiple differences in definition of brain parcels and networks. The corresponding text and figures regarding this additional analysis are included below for the reviewer's benefit:

METHODS:

“As the choice of parcellation can influence network connectivity profiles (39), we furthermore investigated whether the results of this part of the analysis would replicate when using an independently derived set of resting-state networks. We used the set of 300 ROIs and associated network assignments described in (40), which included the 264 ROIs described in (41) with added subcortical and cerebellar ROIs (*Figure S3*). This parcellation used a set of functionally defined spherical ROIs in volumetric space, and employed the Infomap algorithm to compute network assignments for each ROI. As this analysis was intended as a replication of our primary results, we only selected the networks that aligned closely with the CAB-NP networks (see *Table S9* for an overview).” LINES 209-216

“Finally, to test the replicability of the brain model, we used the R package ‘lavaan’ to fit a SEM representing the discovered causal model (CAB-NP) (24) to the alternately derived brain network data (Greene-300) (38).” LINES 260-262

RESULTS:

“Furthermore, the model was highly replicable when using a separately defined brain parcellation (RMSEA = .06, Tucker-Lewis Index = .87, every edge $p < .001$) (Figure S4). As such, the results described here do not appear to depend on the specific parcellation used to measure fMRI resting-state network connectivity.”
 LINES 323-326

SUPPLEMENT:

Figure S1. To determine whether our main results would replicate in an independently-derived brain network parcellation, we used a slightly modified version of the Green-300 ROI-based parcellation described in (1). As this was intended as a replication analysis, we brought these networks into agreement with the networks from our primary analysis by merging the cingulo-opercular and salience networks into one network, and merging the lateral and dorsal somatomotor network into a single somatomotor network.

Table S3. Correspondence between primary analysis brain networks (2) and replication networks (1). Correlations represent the correlation of average within-network connectivity values, across subjects. Networks with no clear correspondence are indicated with a * (not used for replication).

Cole-Anticevic	Greene-300	Correlation
Frontoparietal	Frontoparietal	$r = .84, p = 3.6e-289$
Default Mode	Default Mode	$r = .91, p \approx 0$
Dorsal Attention	Dorsal Attention	$r = .80, p = 4.5e-237$
CinguloOpercular	CinguloOpercular & Salience	$r = .86, p \approx 0$
Auditory	Auditory	$r = .87, p \approx 0$
Visual2	Visual	$r = .93, p \approx 0$
Somatomotor	Somatomotor (Lateral & Dorsal)	$r = .95, p \approx 0$
Language	Ventral Attention	$r = .86, p \approx 0$
Orbitoffective	*	*
Ventral Multimodal	*	*
Posterior Multimodal	*	*
*	SOFA	*
*	Medial Temporal	*
*	Parietomedial	*

Figure S4. We replicated our discovered causal model using a subset of the networks defined in (1). We fit an a priori structural equation model (SEM) to these alternatively defined networks using the edges discovered in our analysis of the Cole-Anticevic networks. Since the Greene-300 parcellation included only one visual network, while the Cole-Anticevic contained two, we merged the discovered visual1 and visual2 edges for the SEM. Specifically, in the discovered graph [Figure 3], visual2 connectivity was caused by somatomotor and dorsal attention connectivity, and visual1 connectivity caused auditory connectivity; in the replication SEM, we included incoming visual edges from somatomotor and dorsal attention, and an outgoing edge to auditory. Standardized edge weights recovered via SEM are displayed in text next to each edge in the graph. The overall SEM fit was almost as good as in the initial parcellation, RMSEA = .06, TLI = .87.

We thank the editor and the reviewers for their careful consideration and thoughtful comments on our manuscript. We look forward to the result of these revisions, which we believe you will find address the concerns of the reviewers satisfactorily.

Sincerely

Eric Rawls, PhD

Erich Kummerfeld, PhD

Anna Zilverstand, PhD

REVIEWERS' COMMENTS:

Reviewer #1 (Remarks to the Author):

The authors responded very thoroughly to my comments and I would like to commend them for their efforts to incorporate my suggestions. I think the revised manuscript has much improved as a result and I am happy to recommend acceptance.

Reviewer #3 (Remarks to the Author):

Thank you for your time and the revisions. I hope that they enhanced your paper in order to maximize its impact in the community. The paper is now in a good shape to be published.

Please find below the final comments from the two reviewers for our manuscript.

REVIEWERS' COMMENTS:

Reviewer #1 (Remarks to the Author):

The authors responded very thoroughly to my comments and I would like to commend them for their efforts to incorporate my suggestions. I think the revised manuscript has much improved as a result and I am happy to recommend acceptance.

Reviewer #3 (Remarks to the Author):

Thank you for your time and the revisions. I hope that they enhanced your paper in order to maximize its impact in the community. The paper is now in a good shape to be published.

We would like to thank the reviewers for their helpful comments and suggestions in maximizing the impact and rigor of our manuscript.

Eric Rawls, PhD

Erich Kummerfeld, PhD

Anna Zilverstand, PhD